# One-second coherence for a single electron spin coupled to a multi-qubit nuclear-spin environment

M.H. Abobeih[1,2], J. Cramer[1,2], M.A. Bakker[1,2], N. Kalb[1,2], M. Markham[3], D.J. Twitchen[3] & T.H. Taminiau[1,2]

Single electron spins coupled to multiple nuclear spins provide promising multi-qubit registers for quantum sensing and quantum networks. The obtainable level of control is determined by how well the electron spin can be selectively coupled to, and decoupled from, the surrounding nuclear spins. Here we realize a coherence time exceeding a second for a single nitrogen-vacancy electron spin through decoupling sequences tailored to its microscopic nuclear-spin environment. First, we use the electron spin to probe the environment, which is accurately described by seven individual and six pairs of coupled carbon-13 spins. We develop initialization, control and readout of the carbon-13 pairs in order to directly reveal their atomic structure. We then exploit this knowledge to store quantum states in the electron spin for over a second by carefully avoiding unwanted interactions. These results provide a proof-of-principle for quantum sensing of complex multi-spin systems and an opportunity for multi-qubit quantum registers with long coherence times.

---

[1] QuTech, Delft University of Technology, PO Box 50462600 GA Delft, The Netherlands. [2] Kavli Institute of Nanoscience Delft, Delft University of Technology, PO Box 50462600 GA Delft, The Netherlands. [3] Element Six Innovation, Fermi Avenue, Harwell Oxford, Didcot, Oxfordshire OX11 0QR, United Kingdom. Correspondence and requests for materials should be addressed to T.H.T. (email: T.H.Taminiau@TUDelft.nl)

Coupled systems of individual electron and nuclear spins in solids are a promising platform for quantum information processing[1–6] and quantum sensing[7–11]. Initial experiments have demonstrated the detection and control of several nuclear spins surrounding individual defect or donor electron spins[12–17]. These nuclear spins provide robust qubits that enable enhanced quantum sensing protocols[7–11], quantum error correction[2,3,18], and multi-qubit nodes for optically connected quantum networks[19–22].

The level of control that can be obtained is determined by the electron spin coherence and therefore by how well the electron can be decoupled from unwanted interactions with its spin environment. Electron coherence times up to 0.56 s for a single electron spin qubit[5] and ∼3 s for ensembles[23–26] have been demonstrated in isotopically purified samples depleted of nuclear spins, but in those cases the individual control of multiple nuclear-spin qubits is forgone.

Here we realize a coherence time exceeding 1 s for a single electron spin in diamond that is coupled to a complex environment of multiple nuclear-spin qubits. First, we use the electron spin as a quantum sensor to probe the microscopic structure of the surrounding nuclear-spin environment, including interactions between the nuclear spins. We find that the spin environment is accurately described by seven isolated single $^{13}$C spins and six pairs of coupled $^{13}$C spins (Fig. 1a). We then develop pulse sequences to initialize, control and readout the state of the $^{13}$C–$^{13}$C pairs. We use this control to directly characterize the coupling strength between the $^{13}$C spins, thus revealing their atomic structure given by the distance between the two $^{13}$C atoms and the angle they make with the magnetic field. Finally, we exploit this extensive knowledge of the microscopic environment to realize tailored decoupling sequences that effectively protect arbitrary quantum states stored in the electron spin for well over a second. This combination of a long electron spin coherence time and selective couplings to a system of up to 19 nuclear spins provides a promising path to multi-qubit registers for quantum sensing and quantum networks.

## Results

**System**. We use a single nitrogen-vacancy (NV) center (Fig. 1a) in a CVD-grown diamond at a temperature of 3.7 K with a natural 1.1% abundance of $^{13}$C and a negligible nitrogen concentration (<5 parts per billion). A static magnetic field of $B_z \approx 403$ G is applied along the NV-axis with a permanent magnet (Methods). The NV electron spin is read out in a single shot with an average fidelity of 95% through spin-selective resonant excitation[27]. The electron spin is controlled using microwave pulses through an on-chip stripline (Methods).

**Longitudinal relaxation**. We first address the longitudinal relaxation ($T_1$) of the NV electron spin, which sets a limit on the maximum coherence time. At 3.7 Kelvin, spin-lattice relaxation due to two-phonon Raman and Orbach-type processes are negligible[28,29]. No cross relaxation to P1 or other NV centers is expected due to the low nitrogen concentration. The electron spin can, however, relax due to microwave noise and laser background introduced by the experimental controls (Fig. 1). We ensure a high on/off ratio of the lasers (>100 dB) and use switches to suppress microwave amplifier noise (see Methods). Figure 1b shows the measured electron spin relaxation for all three initial states. We fit the average fidelity $F$ to

$$F = 2/3\,e^{-t/T_1} + 1/3 \qquad (1)$$

The obtained decay time $T_1$ is $(3.6 \pm 0.3) \times 10^3$ s. This value sets a lower limit for the spin relaxation time, and is the longest

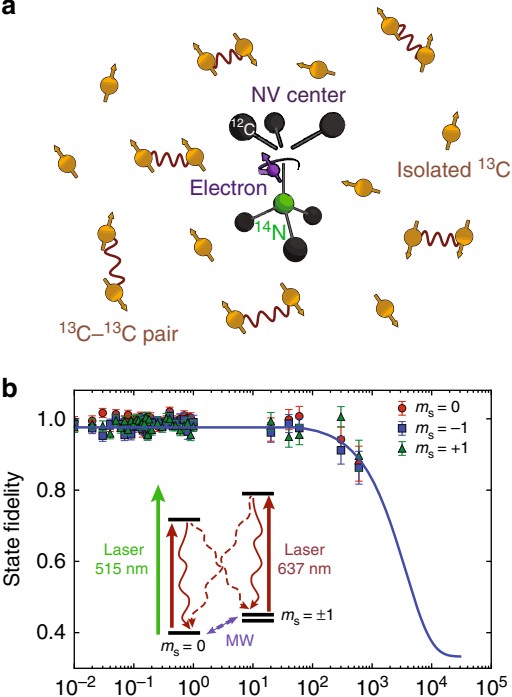

**Fig. 1** Experimental system and $T_1$ measurements. **a** We study a single nitrogen-vacancy (NV) center in diamond surrounded by a bath of $^{13}$C nuclear spins (1.1% abundance). In this work, we show that the microscopic nuclear-spin environment is accurately described by 7 isolated $^{13}$C spins, 6 pairs of coupled $^{13}$C spins and a background bath of $^{13}$C spins (not depicted). **b** Longitudinal relaxation of the NV electron spin. The spin is prepared in $m_s = 0, -1,$ or $+1$ and the fidelity with the initial state is measured after time $t$. The inset shows the microwave (MW) and laser controls for the NV spin and charge states, as well as the pathways for spin relaxation induced by potential background noise from these controls. All error bars are one statistical s.d.

reported for a single electron spin qubit. Remarkably, the observed $T_1$ exceeds recent theoretical predictions based on single-phonon processes by more than an order of magnitude[30,31]. To further investigate the origin of the decay, we prepare $m_s = 0$ and measure the total spin population summed over all three states. The total population decays on a similar timescale ($\sim 3.6 \times 10^3$ s), indicating that the decay is caused by a reduction of the measurement contrast, possibly due to drifts in the optical setup (see Methods), rather than by spin relaxation. This suggests that the spin-relaxation time significantly exceeds the measured $T_1$ value. Nevertheless, the long $T_1$ observed here already indicates that longitudinal relaxation is no longer a limiting factor for NV center coherence.

**Quantum sensing of the microscopic spin environment**. To study the electron spin coherence, we first use the electron spin as a quantum sensor to probe its nuclear-spin environment through dynamical decoupling spectroscopy[12–14]. The electron spin is prepared in a superposition $|x\rangle = (|m_s = 0\rangle + |m_s = -1\rangle)/\sqrt{2}$ and a dynamical decoupling sequence of $N$ $\pi$-pulses of the form $(\tau - \pi - \tau)^N$ is applied. The remaining electron coherence is then measured as a function of the time between the pulses $2\tau$. Loss of electron coherence indicates an interaction with the nuclear-spin environment.

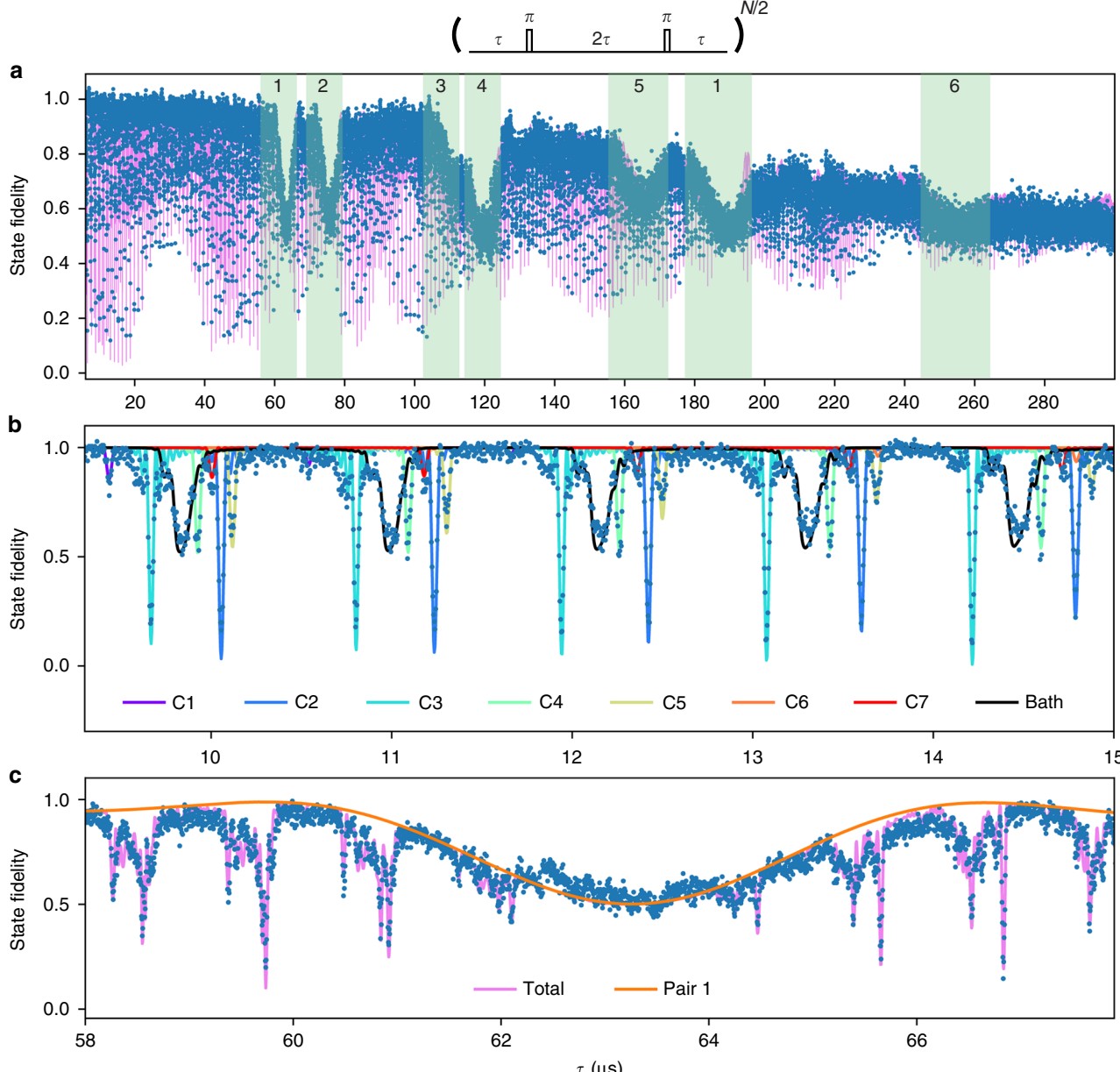

**Fig. 2** Quantum sensing of the microscopic spin environment. **a** Dynamical decoupling spectroscopy[13] revealing a rich nuclear-spin environment consisting of individual $^{13}C$ spins, as well as pairs of coupled $^{13}C$ spins. The electron spin is prepared in a superposition, $|x\rangle = (|m_s = 0\rangle + |m_s = -1\rangle)/\sqrt{2}$ and a decoupling sequence of $N = 32$ $\pi$-pulses separated by $2\tau$ is applied. Loss of coherence indicates the interaction of the electron spin with nuclear spins in the environment. Blue: data. Purple line: theory (see Methods). The shaded areas mark the signals due to six $^{13}C$-$^{13}C$ pairs labeled 1–6. **b** Zoom-in showing sharp signals due to coupling to isolated individual $^{13}C$ spins[12-14]. The total signal is well described by seven $^{13}C$ spins (see Supplementary Table 2 for hyperfine parameters) and a bath of 200 randomly generated spins with hyperfine couplings below 10 kHz. **c** Zoom-in showing a broad signal due to $^{13}C$-$^{13}C$ pair 1[16,32]. Blue: data. The solid orange line is the theoretical signal just due to pair 1, while the purple line includes the seven individual $^{13}C$ spins and the $^{13}C$ spin bath as well

The results in Fig. 2a for $N = 32$ pulses reveal a rich structure consisting of both sharp and broader dips in the electron coherence. The sharp dips (Fig. 2b) have been identified previously as resonances due to the electron spin undergoing an entangling operation with individual isolated $^{13}C$ spins in the environment[12-14]. For this NV center, the observed signal is well explained by seven individual $^{13}C$ spins and a background bath of randomly generated $^{13}C$ spins (Fig. 2b). To verify this explanation we perform direct Ramsey spectroscopy on all seven spins (Supplementary Fig. 1)[3]. For the electron spin in $m_s = \pm 1$, each spin yields a single unique precession frequency due to the

hyperfine coupling, indicating that all seven spins are distinct and do not couple strongly to other $^{13}C$ spins in the vicinity (Supplementary Fig. 1).

The electron can be efficiently decoupled from the interactions with such isolated $^{13}C$ spins by setting $\tau = m \cdot \frac{2\pi}{\omega_L}$, with $m$ a positive integer and $\omega_L$ the $^{13}C$ Larmor frequency for $m_s = 0$[33]. In practice, however, this condition might not be exactly and simultaneously met for all spins due to: the limited timing resolution of $\tau$ (here 1 ns), measurement uncertainty in the value $\omega_L$, and differences between the $m_s = 0$ frequencies for different $^{13}C$ spins, for example caused by different effective g-tensors

under a slightly misaligned magnetic field (here <0.35°, Supplementary Note 3)[3,33–35]. We numerically simulate these deviations from the ideal condition and find that, for our range of parameters, the effect on the electron coherence is small and can be neglected (Supplementary Fig. 2).

We associate the broader dips in Figs. 2a and 2c to pairs of strongly coupled $^{13}$C spins. Such $^{13}$C–$^{13}$C pairs were treated theoretically[32,36] and the signal due to a single pair of nearest-neighbor $^{13}$C spins with particularly strong couplings to a NV center has been detected[16]. In this work, we exploit improved coherence times to detect up to six pairs, including previously undetected non-nearest-neighbor pairs. We then develop pulse sequences to polarize and coherently control these pairs to be able to directly reveal their atomic structure through spectroscopy.

**Direct spectroscopy of nuclear-spin pairs**. The evolution of $^{13}$C–$^{13}$C pairs can be understood from an approximate pseudo-spin model in the subspace spanned by $|\uparrow\downarrow\rangle = |\Uparrow\rangle$ and $|\downarrow\uparrow\rangle = |\Downarrow\rangle$, following Zhao et al.[32] (Supplementary Notes 1 and 2). The pseudo-spin Hamiltonian depends on the electron spin state. For $m_s = 0$ we have:

$$\hat{H}_0 = X\hat{S}_x, \tag{2}$$

and for $m_s = -1$:

$$\hat{H}_1 = X\hat{S}_x + Z\hat{S}_z, \tag{3}$$

where $\hat{S}_x$ and $\hat{S}_z$ are the spin-$\frac{1}{2}$ operators. $X$ is the dipolar coupling between the $^{13}$C spins and $Z$ is due to the hyperfine field gradient (Supplementary Note 2)[32]. The evolution of the $^{13}$C–$^{13}$C pair during a decoupling sequence will thus in general depend on the initial electron spin state, causing a loss of electron coherence.

We now show that this conditional evolution enables direct spectroscopy of the $^{13}$C–$^{13}$C dipolar interaction $X$. Consider two limiting cases: $X \gg Z$ and $Z \gg X$, which cover the pairs observed in this work. In both cases, loss of the electron coherence is expected for the resonance condition $\tau = \tau_k = (2k - 1)\frac{\pi}{2\omega_r}$, with $k$ a positive integer and resonance frequency $\omega_r = \sqrt{X^2 + (Z/2)^2}$[13,32,37]. For $X \gg Z$ the net evolution at resonance is a rotation around the $z$-axis with the rotation direction conditional on the initial electron state (mathematically analogous to the case of a single-$^{13}$C spin in a strong magnetic field[13,38]). For $Z \gg X$ the net evolution is a conditional rotation around the $x$-axis (analogous to the nitrogen nuclear spin subjected to a driving field[37]). These conditional rotations provide the controlled gate operations required to initialize, coherently control and directly probe the pseudo-spin states.

The measurement sequences for the two cases are shown in Fig. 3a. First, a dynamical decoupling sequence is performed that correlates the electron state with the pseudo-spin state. Reading out the electron spin in a single shot then performs a projective measurement that prepares the pseudo-spin into a polarized state. For $X \gg Z$ the pseudo-spin is measured along its $z$-axis and thus prepared in $|\Uparrow\rangle$. For $Z \gg X$ the measurement is along the $x$-axis and the spin is prepared in $(|\Uparrow\rangle + |\Downarrow\rangle)/\sqrt{2}$. Second, we let the pseudo-spin evolve freely with the electron spin in one of its eigenstates ($m_s = 0$ or $m_s = -1$) so that we directly probe the precession frequencies $\omega_0 = X$ (for $m_s = 0$) or $\omega_1 = \sqrt{X^2 + Z^2}$ (for $m_s = -1$). For $Z \gg X$, an extra complication is that the initial state $(|\Uparrow\rangle + |\Downarrow\rangle)/\sqrt{2}$ is an eigenstate of $\hat{H}_0$. To access $\omega_0 = X$, we prepare $(|\Uparrow\rangle + i|\Downarrow\rangle)/\sqrt{2}$ — a superposition of $\hat{H}_0$ eigenstates—by first letting the system evolve under $\hat{H}_1$ for a time $\pi/(2\omega_1)$. Finally, the state of the pseudo-spin is readout through a second measurement sequence.

We find six distinct sets of frequencies (Fig. 3b), indicating that six different $^{13}$C–$^{13}$C pairs are detected. The measurements for $m_s = 0$ directly yield the coupling strengths $X$ and therefore the atomic structure of the pairs (Fig. 4a). We observe a variety of coupling strengths corresponding to nearest-neighbor pairs ($X/2\pi = 2082.7(7)$ Hz, theoretical value 2061 Hz), as well as pairs separated by several bond lengths (e.g., $X/2\pi = 133.8(1)$ Hz, theoretical value 133.4 Hz). The observed number of pairs is consistent with the $^{13}$C concentration of the sample (Supplementary Fig. 4). Note that for pair 4, we have $X \gg Z$, so the resonance condition is mainly governed by the coupling strength $X$. This makes it likely that additional pairs with the same dipolar coupling $X$—but smaller $Z$-values—contribute to the observed signal at $\tau = 120$ μs. Nevertheless, the environment can be described accurately by the six identified pairs, which we verify by comparing the measured dynamical decoupling curves for different values of $N$ to the calculated signal based on the extracted couplings (Fig. 4b).

**Electron spin coherence time**. Next, we exploit the obtained microscopic picture of the nuclear spin environment to investigate the electron spin coherence under dynamical decoupling. To extract the loss of coherence due to the remainder of the dynamics of the environment, i.e., excluding the identified signals from the $^{13}$C spins and pairs, we fit the results to:

$$F = \frac{1}{2} + A \cdot M(t) \cdot e^{-(t/T)^n}, \tag{4}$$

in which $M(t)$ accounts for the signal due to the coupling to the $^{13}$C–$^{13}$C pairs (Fig. 4b, Methods section). $A$, $T$, and $n$ are fit parameters that account for the decay of the envelope due to the rest of the dynamics of the environment and pulse errors. As before, effects of interactions with individual $^{13}$C spins are avoided by setting $\tau = m \cdot \frac{2\pi}{\omega_L}$. An additional challenge is that at high numbers of pulses the electron spin becomes sensitive even to small effects, such as spurious harmonics due to finite MW pulse durations[39,40] and non-secular Hamiltonian terms[41], which cause loss of coherence over narrow ranges of $\tau$ (<10 ns). Here we avoid such effects by scanning a range of ~20 ns around the target value to determine the optimum value of $\tau$.

Figure 5a shows the electron coherence for sequences from $N = 4$ to 10,240 pulses. The coherence times $T$, extracted from the envelopes, reveal that the electron coherence can be greatly extended by increasing the number of pulses $N$. The maximum coherence time is $T = 1.58(7)$ s for $N = 10,240$ (Fig. 5b). We determine the scaling of $T$ with $N$ by fitting to $T_{N=4} \cdot (N/4)^\eta$, with $T_{N=4}$ the coherence time for $N = 4$[23,42–45] which gives $\eta = 0.799(2)$. No saturation of the coherence time $T$ is observed yet, so that longer coherence times are expected to be possible. In our experiments, however, pulse errors become significant at larger $N$, causing a decrease in the amplitude $A$ (Supplementary Fig. 7).

**Protecting arbitrary quantum states**. Finally, we demonstrate that arbitrary quantum states can be stored in the electron spin for well over a second by using decoupling sequences that are tailored to the specific microscopic spin environment (Fig. 5c). For a given storage time, we select $\tau$ and $N$ to maximize the obtained fidelity by avoiding interactions with the characterized $^{13}$C spins and $^{13}$C–$^{13}$C pairs. To assess the ability to protect arbitrary quantum states, we average the storage fidelity over the six cardinal states and do not re-normalize the results. The results show that quantum states are protected with a fidelity above the 2/3 limit of a classical memory for at least 0.995 seconds (using $N = 10,240$ pulses) and up to 1.46 s from interpolation of the results. These are the longest coherence times reported for single

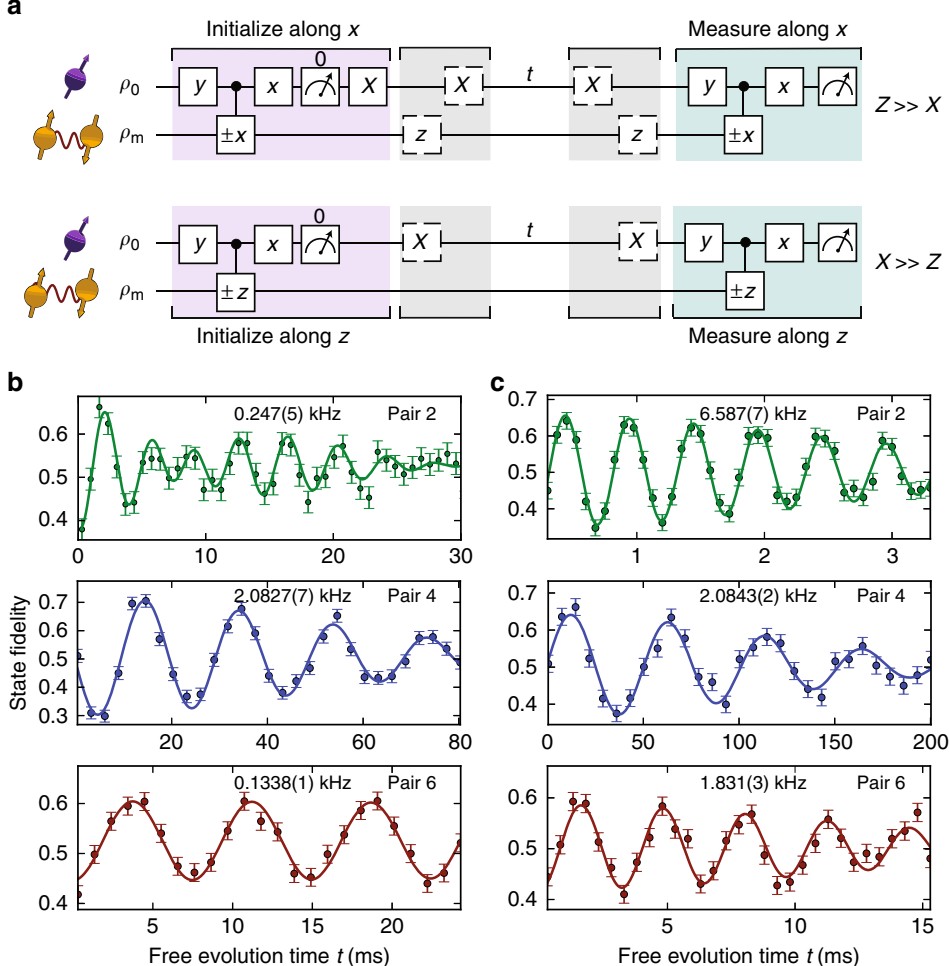

**Fig. 3** Direct spectroscopy of nuclear-spin pairs. **a** Measurement sequences for Ramsey spectroscopy of $^{13}$C–$^{13}$C pairs, for $Z \gg X$ and for $X \gg Z$. The controlled $\pm x$ ($\pm z$) gates are controlled $\pm \pi/2$ rotations around $x$ ($z$) with the sign controlled by the electron. The initial states are $\rho_0 = |0\rangle\langle 0|$ and the mixed state $\rho_m$. **b**, **c** Nuclear spin Ramsey measurements and obtained precession frequencies for pairs 2, 4, and 6. The electron spin state during the free evolution time $t$ is set to $m_s = 0$ (**b**) or $m_s = -1$ (**c**) and an artificial detuning is applied. Each pair yields a unique set of frequencies, confirming that the pairs are distinct. For pair 2 an additional beating is observed (frequency of 23(3) Hz), indicating a small coupling to one (or more) additional spins. See Supplementary Fig. 3 for the other three pairs and Supplementary Table 4 for fit results. All error bars are one statistical s.d.

solid-state electron spin qubits[5], despite the presence of a dense nuclear spin environment that provides multiple qubits.

## Discussion

These results provide opportunities for quantum sensing and quantum information processing, and are applicable to a wide variety of solid-state spin systems[4,5,17,46–56]. First, these experiments are a proof-of-principle for resolving the microscopic structure of multi-spin systems, including the interactions between spins[32]. The developed methods might be applied to detect and control spin interactions in samples external to the host material[10,57–59]. Second, the combination of long coherence times and selective control in an electron-nuclear system containing up to twenty spins enables improved multi-qubit quantum registers for quantum networks. The electron spin coherence now exceeds the time needed to entangle remote NV centers through a photonic link, making deterministic entanglement delivery possible[60]. Moreover, the realized control over multiple $^{13}$C–$^{13}$C pairs provides promising multi-qubit quantum memories with long coherence times, as the pseudo-spin naturally forms a decoherence-protected subspace[61].

## Methods

**Setup**. The experiments are performed at 3.7 K (Montana Cryostation) with a magnetic field of ~403 G applied along the NV-axis by a permanent magnet. We realize long relaxation ($T_1 > 1$ h) and coherence times (>1 s) in combination with fast spin operations (Rabi frequency of 14 MHz) and readout/initialization (~10 μs), by minimizing noise and background from the microwave (MW) and optical controls. Amplifier (AR 25S1G6) noise is suppressed by a fast microwave switch (TriQuint TGS2355-SM) with a suppression ratio of 40 dB. Video leakage noise generated by the switch is filtered with a high pass filter. We use Hermite pulse envelopes[62,63] to obtain effective MW pulses without initialization of the intrinsic $^{14}$N nuclear spin. To mitigate pulse errors we alternate the phases of the pulses following the XY8 scheme[64]. Laser pulses are generated by direct current modulation (515 nm laser, Cobolt MLD - for charge state control) or by acoustic optical modulators (637 nm Toptica DL Pro and New Focus TLB-6704-P for spin pumping and single-shot readout[27]). The direct current modulation yields an on/off ratio of >135 dB. By placing two modulators in series (Gooch and Housego Fibre Q) an on/off ratio of >100 dB is obtained for the 637 nm lasers. The laser frequencies are stabilized to within 2 MHz using a wavemeter (HF-ANGSTROM WS/U-10U). Possible explanations for the observed decay in Fig. 1b are frequency drifts of this wavemeter or spatial drifts of the laser focus over 1-h timescales.

**Sample**. We use a naturally occurring NV center in high-purity type IIa homo-epitaxially chemical-vapor-deposition (CVD) grown diamond with a 1.1% natural abundance of $^{13}$C and a ⟨111⟩ crystal orientation (Element Six). The NV center studied here has been selected for the absence of very-close-by strongly coupled $^{13}$C spins (>500 kHz hyperfine coupling), but not on any other properties of the nuclear spin environment. To enhance the collection efficiency a solid-immersion

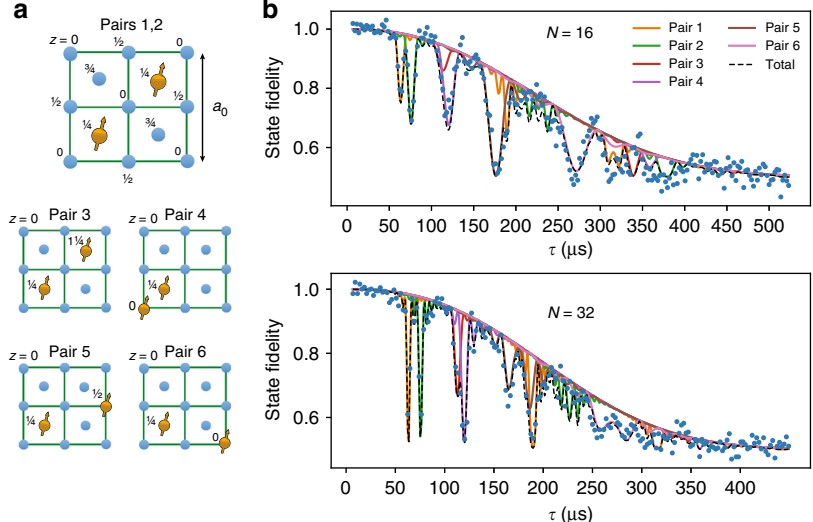

**Fig. 4** Atomic structure and decoupling signal for the six nuclear-spin pairs. **a** Structure of the six $^{13}C$-$^{13}C$ pairs within the diamond unit cell (up to symmetries and equivalent orientations). The $z$-values give the height in fractions of the diamond lattice constant $a_0$. The magnetic field is oriented along the <111> direction, i.e., along the axis of pair 4. For pair 3 there is an additional possible structure that yields a similar $X$, Supplementary Table 3. **b** The calculated signal for the six individual $^{13}C$-$^{13}C$ pairs accurately describes the measured decoupling signal for different number of pulses $N$. Data are taken for $\tau = m \cdot \frac{2\pi}{\omega_L}$ to avoid coupling to single-$^{13}C$ spins. See Supplementary Fig. 5 for other values of $N$

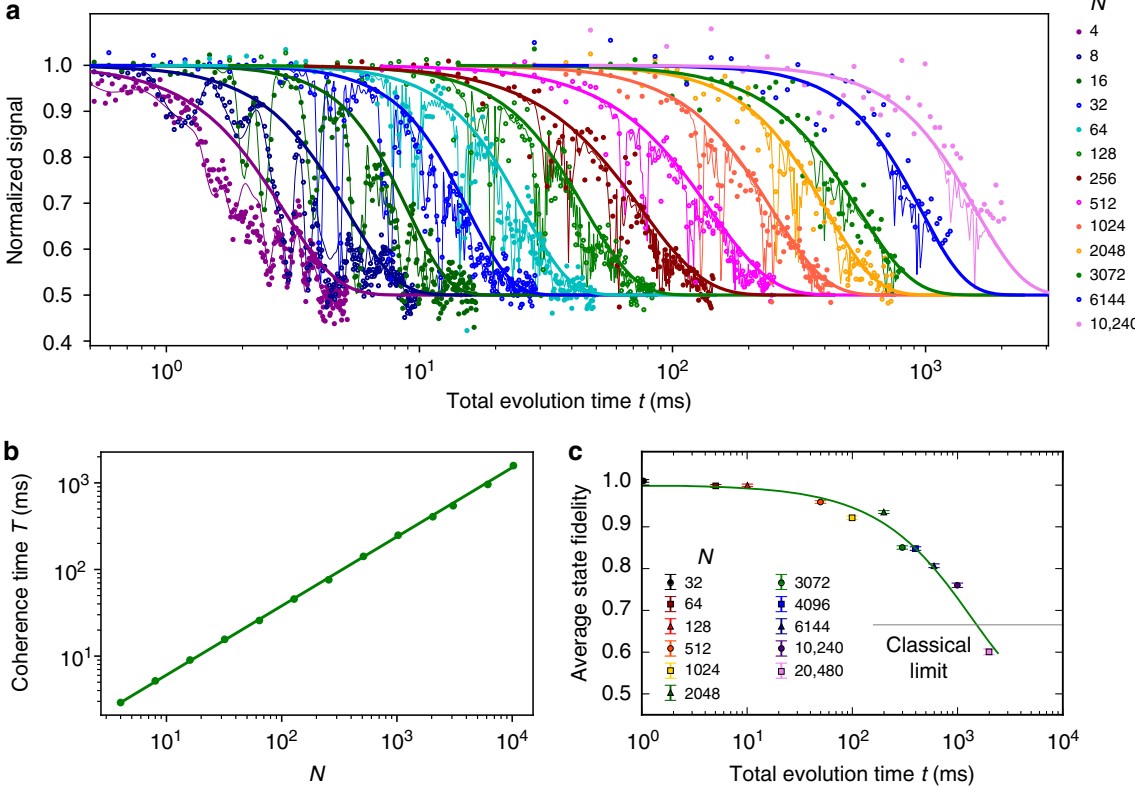

**Fig. 5** Protecting quantum states with tailored decoupling sequences. **a** Normalized signal under dynamical decoupling with the number of pulses varying from $N = 4$ to $N = 10,240$. The electron is initialized and readout along $x$. The thin lines are fits to equation (4), which takes into account the six identified $^{13}C$-$^{13}C$ pairs. We use the extracted amplitudes $A$ to re-normalize the signal. Thick lines are the extracted envelops $\left(0.5 + 0.5 \cdot e^{-(t/T)^n}\right)$ with $T$ and $n$ obtained from the fits. See Supplementary Fig. 6 for the obtained values $n$. **b** Scaling of the obtained coherence time $T$ as function of the number of pulses (error bars are <5%). The solid line is a fit to the power function $T_{N=4} \cdot (N/4)^\eta$, where $T_{N=4}$ is the coherence time for $N = 4$. We find $\eta = 0.799(2)$. **c** The average state fidelity obtained for the six cardinal states (Supplementary Fig. 8). Unlike in **a**, the signal is shown without any renormalization. The number of pulses $N$ is chosen to maximize the obtained signal at the given total evolution time while avoiding interactions with the $^{13}C$ environment. The solid green line is a fit to an exponential decay. The horizontal line at $\frac{2}{3}$ fidelity marks the classical limit for storing quantum states. The two curves cross at $t = 1.46$ s demonstrating the protection of arbitrary quantum states well beyond a second. All error bars are one statistical s.d.

lens was fabricated on top of the NV center[27,65] and a single-layer aluminum-oxide anti-reflection coating was deposited[66,67].

**Data analysis.** We describe the total signal for the NV electron spin after a decoupling sequence in Fig. 2 as:

$$F = \frac{1}{2} + A \cdot M_{\text{bath}}(t) \cdot \prod_{i=1}^{7} M_{\text{C}}^{i}(t) \cdot \prod_{j=1}^{6} M_{\text{pair}}^{j}(t) \cdot e^{-(t/T)^{n}}, \qquad (5)$$

where $t$ is the total time. $M_{\text{bath}}$ is the signal due to a randomly generated background bath of non-interacting spins with hyperfine couplings below 10 kHz. $M_{\text{C}}^{i}$ are the signals due to the seven individual isolated $^{13}$C spins[13]. $M_{\text{pair}}^{j}$ are the signals due to the six $^{13}$C–$^{13}$C pairs and are given by $1/2 + \text{Re}(\text{Tr}(U_0 U_1^\dagger))/4$, with $U_0$ and $U_1$ the evolution operators of the pseudo-spin pair for the decoupling sequence conditional on the initial electron state ($m_s = 0$ or $m_s = -1$)[32]. The coherence time $T$ and exponent $n$ describe the decoherence due to remainder of the dynamics of the spin environment.

Setting $\tau = m \cdot 2\pi/\omega_L$ avoids the resonances due to individual $^{13}$C spins, so that equation (5) reduces to:

$$F = \frac{1}{2} + A \cdot \prod_{j=1}^{6} M_{\text{pair}}^{j}(t) \cdot e^{-(t/T)^{n}}. \qquad (6)$$

The data in Figs. 3 and 4 are fitted to equation (6) and $A$, $T$ and $n$ are extracted from these fits.

**Data availability.** The data that support the findings of this study are available from the corresponding author upon request.

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

## Acknowledgements

We thank V. V. Dobrovitski, J. E. Lang, T. S. Monteiro, H. P. Bartling, C. L. Degen, and R. Hanson for valuable discussions, P. Vinke, R. Vermeulen, R. Schouten, and M. Eschen for help with the experimental apparatus, and A. J. Stolk for characterization measurements. We acknowledge support from the Netherlands Organization for Scientific Research (NWO) through a Vidi grant.

## Author contributions

M.H.A. and T.H.T. devised the experiments. M.H.A., J.C., and T.H.T. constructed the experimental apparatus. M.M. and D.J.T. grew the diamond. M.H.A. performed the experiments with support from M.A.B. and N.K. M.H.A. and T.H.T. analyzed the data with help of all authors. T.H.T. supervised the project.

## Additional information

**Competing interests:** The authors declare no competing interests.

