## [Peer Review File · Nature Communications]

Reviewers' comments:

Reviewer #1 (Remarks to the Author):

This paper presents a really impressive combination of experimental control with a careful analysis of the data of a coupled system of one electron spin and 13 nuclear (^{13}C) spins. While the basics of the applied techniques (experimental as well as theoretical) are known, the present paper can be considered an excellent demonstration of the capabilities of these techniques if they are used by such a dedicated group of scientists. I strongly recommend that this paper be published.

I only have minor suggestions for revisions, basically for improving the clarity of the presentation in a few spots:

- The authors stress the "sensing" potential of their technique. However, the paper leads one to the conclusion that the sensing is more a means than a goal: characterising the system is a precondition for achieving the control in such detail.
- The mention that drift is a limiting factor; if possible, it would be interesting to read what the dominant effect of drift is.
- The initial condition is given as " $|x\rangle = (|m_s = 0\rangle + |m_s = -1\rangle)$ " in the text and " $|x\rangle = (|m_s = 0\rangle \pm |m_s = -1\rangle)$ " in the figure caption; please check.
- It would be useful to introduce new quantities with a "name", to help distinguish them from each other. Examples: X, Z, omega, omega_0, omega_1.
- What is meant with "Pauli spin operators" : are the eigenvalues $\pm 1/2$ or ± 1 ?
- "interactions with individual ^{13}C spins are avoided" : I would assume that the interactions are there, but they do not affect the signal?
- "different bare Larmor precession frequencies, ω_0 , depending on their hyperfine coupling parameters" : most people would assume that the term "bare spin" implies that there are not hyperfine coupling parameters
- Figure S2 : - label y-axis
- "The dashed blue line marks the $1/e$ decay times for different values of N " : why is the time shorter for larger N?

Reviewer #2 (Remarks to the Author):

In this paper, the authors studied the structure local environment of a single NV center in diamond by dynamical decoupling techniques. With the obtained knowledge, the authors demonstrated several important applications including control of ^{13}C pairs, and protection of electron spin states for a second. The authors integrated several up-to-date techniques, e.g., single shot readout and many-pulse dynamical decoupling, and set new records for long measured T_1 time of single NVs, identification of 7 individual and 6 pairs of ^{13}C nuclei, and over 1 second long coherence time of NV in diamond with natural abundance ^{13}C . All these results are definitely important for the NV community and will have broad impact in defect-center-based quantum computing and quantum sensing. Therefore, I recommend the publication of this manuscript in Nature Communications, if the following issues are clarified:

1. How accurate can the local environment of the single NV can be determined? In order to make full use of the ^{13}C nuclear spins near the NV center, one needs the precise information about the coupling constants (hyperfine interaction tensor). In the Supplementary Table II., the authors listed the parameters of the detected 7 ^{13}C nuclear spins, but only mentioned that "Uncertainties are estimated to be of the order of the last digit". Can the authors show more data about the uncertainties? As a related problem, if I understand correctly, these parameters are obtained from fitting the data with the pseudo-spin models; I am wondering the confidence level of the fitting process. The authors should give more details about the fitting process, as it is difficult to judge only from the large amount of data shown in Figs. 2a and 2b.

2. The typicality of the results shown in this manuscript. I am wondering whether the number of C13 appearing close to the NV (i.e., 7 individual and 6 pairs) is consistent with the sample condition (i.e. natural abundance 1.1% C13). I believe the authors must have done similar measurements on different NVs. How many C13 can be identified in those cases? The authors should at least comment on that the reported results are the best case, or a typical case we can do using natural abundance diamond.

Reviewer #3 (Remarks to the Author):

In this paper, the authors realized a coherence time exceeding a second for a single electron spin in nuclear spin bath at 3.7 K through decoupling techniques. The authors claim that they "first" used the electron spin to probe the environment, which is accurately described by seven individual and six pairs of coupled ¹³C spins. The authors claim that they exploited the knowledge to store quantum states for over a second by carefully avoiding unwanted interactions.

I agree the authors' claim that the measured T₂ (> a second) is the longest coherence time reported for single solid-state electron spin qubits as far as I know, despite the presence of a dense nuclear spin environment that provides multiple qubits. I admit that the authors showed one of the excellent characteristic and potential of the NV center.

However, I have comments as follows.

(1) The authors realized a long coherence time by decoupling technique at 3.7 K. As far as I see, the essential point to realize it by authors is just by lowering temperature. In reference 23, T₂ = 0.6 second is reported at 77K. It is shown that T₂ is limited by T₁, therefore, they showed T₂ can be extended by lowering temperature and decoupling techniques. The fig 5b in the present paper looks like lower temperature version of figure 2b in reference 23.

Therefore, I can not admit the scientific advances over the previous researches. I think the authors should explain the advances if they have.

(2) The authors described that "... decoupling sequences tailored to its microscopic nuclear-spin environment ..." and "we then exploit this knowledge to store quantum states by carefully avoiding unwanted interactions."

It is true that the authors analyzed deeply about the nuclear spin environment. However, the authors just utilized Hermite pulse shapes for the decoupling pulses described in references 59, 60. Therefore, I feel that the above expressions exaggerated and I can not find the significance of the technical advances over the previous researches. I think the authors should explain the advances if they have.

Other comments:

(3) In the abstract, the authors claim that they "first" used the electron spin to probe the environment, ..., however I do not think so because there are many researches to probe magnetic environments by using the electron spin of the NV center. I think it should be revised.

(4) In page 9, the authors describe that pulse errors become the limiting factor at larger N, causing a decrease in the amplitude A. I think the authors should show the evidence.

Reviewer #1 (Remarks to the Author):

This paper presents a really impressive combination of experimental control with a careful analysis of the data of a coupled system of one electron spin and 13 nuclear (^{13}C) spins. While the basics of the applied techniques (experimental as well as theoretical) are known, the present paper can be considered an excellent demonstration of the capabilities of these techniques if they are used by such a dedicated group of scientists. I strongly recommend that this paper be published. I only have minor suggestions for revisions, basically for improving the clarity of the presentation in a few spots:

We thank the reviewer for the positive assessment of our work and the endorsement for publication in Nature Communications.

- *The authors stress the "sensing" potential of their technique. However, the paper leads one to the conclusion that the sensing is more a means than a goal: characterising the system is a precondition for achieving the control in such detail.*

Indeed, in this work the sensing of the nuclear spin environment mainly serves as a means to realize improved control and coherence. Nevertheless, the presented methods are more general. For example, the measurements in Figure 3 can in principle be used to sense and quantitatively characterize interactions in external spin samples. We think it is important to include this alternative point of view and the connection of our work to the relevant quantum sensing literature.

Note that in the introduction we also discuss a second connection of our work to quantum sensing: the nuclear spins can be used as memories or qubits to enable enhanced sensing techniques (see e.g. refs 7 to 11).

- The mention that drift is a limiting factor; if possible, it would be interesting to read what the dominant effect of drift is.

The decay of the T_1 signal in Fig. 1b on one-hour timescales is expected to be due to drift of the optical setup, which causes a (small) reduction of the readout contrast. Likely candidates are frequency drifts of the wavemeter used to stabilize the laser frequency (due to temperature and pressure fluctuations) and/or slow spatial drifts of the optical focus relative to the NV center. Both these effects would reduce the number of observed photons and thus the readout contrast. We have added a brief description of these possible explanations for the decay in the methods section.

- The initial condition is given as " $|x\rangle = (|m_s = 0\rangle + |m_s = -1\rangle)$ " in the text and " $|x\rangle = (|m_s = 0\rangle \pm |m_s = -1\rangle)$ " in the figure caption; please check.

We thank the reviewer for catching this. We have corrected the state in the figure caption to $|x\rangle = (|m_s = 0\rangle + |m_s = -1\rangle)$.

- It would be useful to introduce new quantities with a "name", to help distinguish them from each other. Examples: X, Z, ω , ω_0 , ω_1 .

Where possible, we have now added descriptive names when these quantities appear (e.g. "the dipolar coupling X"). Additionally we have renamed "omega" to "the resonance frequency ω_r ".

- What is meant with "Pauli spin operators" : are the eigenvalues $\pm 1/2$ or ± 1 ?

We have changed the naming to “spin-1/2 operators” (eigenvalues $\pm \frac{1}{2}$), in order to avoid confusion with the Pauli matrices (eigenvalues ± 1).

- "interactions with individual ^{13}C spins are avoided" : I would assume that the interactions are there, but they do not affect the signal?

We agree with the reviewer that the interactions are present, but that we avoid their effects by canceling them. We have clarified the manuscript to “effects of interactions with individual ^{13}C spins are avoided”.

- "different bare Larmor precession frequencies, ω_0 , depending on their hyperfine coupling parameters" : most people would assume that the term "bare spin" implies that there are not hyperfine coupling parameters

We removed the word “bare” throughout the paper and supplementary information in order to clarify that ω_0 is simply the precession frequency for the electron in spin state $m_s=0$, which can be affected by the hyperfine coupling through misaligned magnetic fields (as explained in Supplementary Note 3).

- Figure S2 : - label y-axis

We have now added y-axis labels (previously the label was in the title of the figure).

- "The dashed blue line marks the $1/e$ decay times for different values of N " : why is the time shorter for larger N ?

The time on x-axis of Supplementary Fig. 2 is half of the time between the pulses (τ) and not the total time. Because the coherence time does not scale linearly with the number of pulses but rather as $N^{0.799}$ (see Fig. 5b), the $1/e$ value of τ becomes shorter for larger N (although the total coherence time, $2*N*\tau$, increases). We have added a note in the figure caption to clarify this.

Reviewer #2 (Remarks to the Author):

In this paper, the authors studied the structure local environment of a single NV center in diamond by dynamical decoupling techniques. With the obtained knowledge, the authors demonstrated several important applications including control of ^{13}C pairs, and protection of electron spin states for a second. The authors integrated several up-to-date techniques, e.g., single shot readout and many-pulse dynamical decoupling, and set new records for long measured T_1 time of single NVs, identification 7 individual and 6 pairs of ^{13}C nuclei, and over 1 second long coherence time of NV in diamond with natural abundance ^{13}C . All these results are definitely important for the NV community and will have broad impact in defect-center-based quantum computing and quantum sensing. Therefore, I recommend the publication of this manuscript in Nature Communications, if the following issues are clarified:

We thank the referee for the careful assessment of our manuscript and for the recommendation for publication in Nature Communications.

1. How accurate can the local environment of the single NV can be determined? In order to make full use of the ^{13}C nuclear spins near the NV center, one needs the precise information about the coupling constants (hyperfine interaction tensor). In the Supplementary Table II., the authors listed

the parameters of the detected 7 C13 nuclear spins, but only mentioned that “Uncertainties are estimated to be of the order of the last digit”. Can the authors show more data about the uncertainties? As a related problem, if I understand correctly, these parameters are obtained from fitting the data with the pseudo-spin models; I am wondering the confidence level of the fitting process. The authors should give more details about the fitting process, as it is difficult to judge only from the large amount of data shown in Figs. 2a and 2b.

Our measurements enable us to accurately determine the precession frequencies ω_0 and ω_1 , whose values and uncertainties are obtained from least-squares fits in Supplementary Fig. 1 (Fig. 3 for 13C pairs). These frequencies are the important parameters for this work, because they enable us to avoid interactions with the 13C spins.

Determining the underlying hyperfine components with a similar accuracy in a self-consistent manner is a non-trivial problem. One would need to account for additional unknown Hamiltonian terms, such as the magnetic field vector, the electron spin g-tensor and the zero-field splitting. Additionally, signals like Fig. 2 are difficult to fit rigorously because they are described by a non-linear function involving a large unknown number of nuclear spins. Therefore, although an interesting problem in itself, accurately determining the hyperfine components (and simultaneously all other required Hamiltonian terms) goes beyond the current manuscript. Instead we report an estimate with conservative uncertainties based on matching the calculated signals for the 7 spins to the signal of Fig. 2a.

To clarify the above we have added to the caption of Supplementary Table II that:

- ω_0 , ω_1 and T_2^* are obtained from least-squares fits of Supplementary Fig. 1.
- the hyperfine components are estimates from Fig. 2.

2. The typicality of the results shown in this manuscript. I am wondering whether the number of C13 appearing close to the NV (i.e., 7 individual and 6 pairs) is consistent with the sample condition (i.e. natural abundance 1.1% C13). I believe the authors must have done similar measurements on different NVs. How many C13 can be identified in those cases? The authors should at least comment on that the reported results are the best case, or a typical case we can do using natural abundance diamond.

The number of spins observed is expected to be typical. The NV center studied here has been selected only for the absence of very-close-by strongly coupled C13 spins (>500 kHz hyperfine), but not on any other properties of the spin environment. Therefore for the purpose of this paper the NV provides a randomly selected sample. We now added this information to the method section.

For the individual C13 spins, the number of observed spins is similar to previously studied NV centers, see for example reference 13 and Kalb et al. *arXiv:1802.05996* (2018). Note that although each NV is unique, the relatively large number of observed spins means that variations are expected to be relatively small (for the closest 700 carbon atoms there are on average ~ 7 C13 spins with a standard deviation of ~ 2.6).

For the C13 pairs, our work provides the first experimental data of this type. To show that the number of observed pairs is within expectations, we now provide a numerical calculation of the number of pairs with dipole- and hyperfine-coupling parameters in the range of the observed pairs (see new Supplementary Figure 4). The results for 10000 randomly generated NV centers yield an average of 9.4 pairs with these properties and a standard deviation of 3.7, showing that the observed number of pairs is consistent with the C13 concentration. Additionally, we added a sentence summarizing this conclusion in the main text.

Reviewer #3 (Remarks to the Author):

In this paper, the authors realized a coherence time exceeding a second for a single electron spin in nuclear spin bath at 3.7 K through decoupling techniques. The authors claim that they "first" used the electron spin to probe the environment, which is accurately described by seven individual and six pairs of coupled ^{13}C spins. The authors claim that they exploited the knowledge to store quantum states for over a second by carefully avoiding unwanted interactions.

I agree the authors' claim that the measured T_2 (> a second) is the longest coherence time reported for single solid-state electron spin qubits as far as I know, despite the presence of a dense nuclear spin environment that provides multiple qubits. I admit that the authors showed one of the excellent characteristic and potential of the NV center.

We thank the reviewer for their report and for highlighting the potential of the record coherence times combined with access to multiple qubits that we present in our manuscript.

However, I have comments as follows.

(1) The authors realized a long coherence time by decoupling technique at 3.7 K. As far as I see, the essential point to realize it by authors is just by lowering temperature. In reference 23, $T_2 = 0.6$ second is reported at 77K. It is shown that T_2 is limited by T_1 , therefore, they showed T_2 can be extended by lowering temperature and decoupling techniques.

The fig 5b in the present paper looks like lower temperature version of figure 2b in reference 23. Therefore, I can not admit the scientific advances over the previous researches. I think the authors should explain the advances if they have.

First, let us stress again that our work is performed on a single NV center surrounded by nuclear spin qubits. Our main result is the long (>1 second) coherence for the NV center in combination with the presence and control of many nuclear spin qubits. In stark contrast, reference 23 is performed on an ensemble of NV centers (no single-spin control) and in an isotopically purified sample. As we explain in the introduction, these conditions make the desired control of many nuclear-spin qubits impossible.

Second, the reviewer argues that in reference 23 the coherence time (T_2) is limited by temperature and that longer coherence times could thus have been obtained by simply lowering the temperature. This is incorrect: the coherence time in reference 23 is not limited by temperature. T_1 exceeds 10 seconds at 77K, as is clearly stated at the bottom left of page 3 in reference 23, whereas the maximum reported coherence time is 0.58 seconds. Our Fig. 5b is thus not "a lower temperature version" of Fig. 2b in reference 23 and temperature plays no direct role in either result. Instead, our results probe completely different physics: decoherence due to a nuclear spin bath, after taking into account the microscopic structure and coherent interactions that we reveal in this work.

The essence of our work is therefore not a lower temperature. Instead, our key advance is the careful probing and subsequent decoupling of the interactions of the NV with its microscopic nuclear spin environment. In this way we realize a record-long coherence time for single-electron-spin qubits, while maintaining access to multiple nuclear spin qubits.

(2) The authors described that "... decoupling sequences tailored to its microscopic nuclear-spin environment ... " and "we then exploit this knowledge to store quantum states by carefully avoiding unwanted interactions."

It is true that the authors analyzed deeply about the nuclear spin environment. However, the authors just utilized Hermite pulse shapes for the decoupling pulses described in references 59, 60. Therefore, I feel that the above expressions exaggerated and I can not find the significance of the technical advances over the previous researches. I think the authors should explain the advances if they have.

The advance in our work is that in Fig. 5C we decouple unwanted interactions with the nuclear spin environment by carefully choosing the number of pulses N and the time τ between these pulses, based on our microscopic characterization of the spin environment.

These parameters (N and τ) should not be confused with the shapes of the microwave pulses themselves. As mentioned in the methods, we use Hermite pulses to realize high-fidelity pulses on the NV center without having to initialize the ^{14}N spin of the NV. Other than this somewhat technical detail, the Hermite pulse shapes play little to no role on decoupling the NV center from the ^{13}C nuclear spin environment, which is the topic of this paper.

Other comments:

(3) In the abstract, the authors claim that they "first" used the electron spin to probe the environment, ..., however I do not think so because there are many researches to probe magnetic environments by using the electron spin of the NV center. I think it should be revised.

This seems to be a misunderstanding related to English grammar. We intended the word "first" to indicate the order of the steps/experiments that we report ("First... Then..."). It is not a claim of novelty that we are the first to use the electron spin to probe magnetic environments ("... for the first time ...").

To ensure that there cannot be any confusion, we changed the sentence to "First, we use the electron spin to probe the environment." and also corrected the same construction in the introduction.

(4) In page 9, the authors describe that pulse errors become the limiting factor at larger N , causing a decrease in the amplitude A . I think the authors should show the evidence.

We thank the referee for this comment. Indirect evidence was already shown and discussed in Supplementary Fig. 6 (now Sup. Fig. 8), but this was indeed not referenced as such in the main text. To show direct evidence of the decrease in the amplitude A , we now provide the data of Fig. 5A without normalization in a new Supplementary Fig. 7.

Additionally, we have slightly adapted the sentence on page 9 to make clearer that, for large N , pulse errors limit the experiments in terms of obtainable amplitude (signal-to-noise ratio and fidelity), not coherence time.

Other changes made upon revision:

- Added the following related references on page 4:
(34) Stanwix, P. L. et al. Coherence of nitrogen-vacancy electronic spin ensembles in diamond. Phys. Rev. B 82, 201201 (2010).

(35) Maze, J. R et al. Electron spin decoherence of single nitrogen-vacancy defects in diamond. *Phys. Rev. B* 78, 094303 (2008).

- Added a previously missing type of C13 pair, vector [3,3,3], to the theoretical coupling strengths (Supplementary table IV). This type of pair yields a coupling strength very close to the [2,2,4] pairs. As a result the experimentally observed pair 3 can be attributed either to [3,3,3] or [2,2,4]. We added brief statements clarifying this at the relevant locations in the manuscript (Caption Fig. 4, Supplementary Tables III and IV, and Supplementary note II).
- Corrected two parameters in Supplementary Table II from -11 kHz and 55 kHz to -11.4 kHz and 59 kHz.

REVIEWERS' COMMENTS:

Reviewer #2 (Remarks to the Author):

In the revised version, my previous concerns were fully addressed. I recommend the publication of this manuscript in Nature Communications.

Reviewer #3 (Remarks to the Author):

I read the authors' replies. The authors replied to all my comments. I think the manuscript can be published in Nature Communications.